# Falcon gut microbiota is shaped by diet and enriched in *Salmonella*

**Anique R. Ahmad**[1], **Samuel Ridgeway**[2], **Ahmed A. Shibl**[1], **Youssef Idaghdour**[2], **Aashish R. Jha**[1]*

**1** Genetic Heritage Group, Program in Biology, New York University Abu Dhabi, Abu Dhabi, UAE, **2** Program in Biology, New York University Abu Dhabi, Abu Dhabi, UAE

* jhaar@nyu.edu

**Data Availability Statement:** A phyloseq object containing the 16S data and metadata as well as the analyses protocols used in this work are included in the Supplementary Data. Sequence

## Abstract

The gut microbiome is increasingly being appreciated as a master regulator of animal health. However, avian gut microbiome studies commonly focus on birds of economic importance and the gut microbiomes of raptors remain underexplored. Here we examine the gut microbiota of 29 captive falcons—raptors of historic importance—in the context of avian evolution by sequencing the V4 region of the 16S rRNA gene. Our results reveal that evolutionary histories and diet are significantly associated with avian gut microbiota in general, whereas diet plays a major role in shaping the falcon gut microbiota. Multiple analyses revealed that gut microbial diversity, composition, and relative abundance of key diet-discriminating bacterial genera in the falcon gut closely resemble those of carnivorous raptors rather than those of their closest phylogenetic relatives. Furthermore, the falcon microbiota is dominated by Firmicutes and contains *Salmonella* at appreciable levels. *Salmonella* presence was associated with altered functional capacity of the falcon gut microbiota as its abundance is associated with depletion of multiple predicted metabolic pathways involved in protein mass buildup, muscle maintenance, and enrichment of anti-microbial compound degradation, thus increasing the pathogenic potential of the falcon gut. Our results point to the necessity of screening for *Salmonella* and other human pathogens in captive birds to safeguard both the health of falcons and individuals who come in contact with these birds.

## Introduction

Animals harbor complex collections of symbiotic microorganisms including bacteria, archaea, viruses, and fungi that are collectively known as the microbiota [1]. Emerging evidence indicates that these microbes provide their hosts with additional functions that animals have not yet been able to evolve, allowing them to fill novel ecological niches. Particularly important are the microbes residing in the gut as they can potentially regulate animal immunity, development, metabolism, behavior, and overall health [2–7]. The mammalian gut microbiota is strongly influenced by host evolutionary histories, diet, and environment [8–11] but how these factors contribute to the gut microbiota in birds remains unresolved [12, 13]. Over 10,000 species of birds are distributed worldwide and they exhibit pronounced diversity in

data is available in the GenBank SRA archive
(PRJNA903547).

**Funding:** This study was supported by NYUAD in
the form of a grant awarded to ARJ (ADHPG
AD318) and grant awarded to YI (ADHPG AD105).
The specific roles of these authors are articulated in
the 'author contributions' section. The funders had
no role in study design, data collection and
analysis, decision to publish, or preparation of the
manuscript.

**Competing interests:** The authors have declared
that no competing interests exist.

morphological traits, physiological functions, and adaptations to ecological niches [14]. Consequently, the bird gut microbiota harbors hundreds of bacterial species that play an important role in regulating physiological functions in birds [15]. Because of the important role that gut microbes may play in maintaining avian health, it is essential to identify factors that contribute towards shaping the gut microbiota in birds.

The gut microbiota can be considered an adaptive trait that is regulated by host genetics and environment, particularly diet [12]. Although recent studies have revealed flight, body size, and migration patterns are associated with the gut microbiota in birds [16] the effect of evolutionary histories and diet on the bird gut microbiota remains poorly characterized. Furthermore, studies investigating gut microbiomes of uber-carnivorous raptors whose diet almost exclusively consists of animal flesh, are underrepresented, with the exception of a few studies that include a small number of raptors [16–24]. Birds of the genus *Falco* (henceforth referred to as falcons) are unique among raptors as they are genetically distinct from other flesh-eating birds of prey such as hawks, eagles and vultures, all of which belong to a single taxonomic order–*Accipitriformes* [25]. Falcons' closest genetic relatives are parrots (Order: *Psittaciformes*) and songbirds (Order: *Passeriformes*), with whom they shared a common ancestor approximately 60 million years ago [25, 26]. Given their unique evolutionary history and diet, understanding the factors shaping the gut microbiota in the falcons may allow us to dissect the contributions of evolutionary histories and carnivorous diet on the avian gut microbiota.

In addition to their evolutionary importance, falcons hold a special place in human history and have been used as religious, royal, and national symbols spanning multiple civilizations across several millennia [26, 27]. Falconry, the art of using falcons to hunt birds and small mammals, preserves falcons' status as cultural icons in many Arabian Gulf nations, although falconry as sport is gaining popularity across the world [28]. The interactions between humans and falcons during falconry may provide opportunities for zoonotic transfers of potential pathogens in the falcon gut that can cause deadly human diseases [29]. Although whether falcons harbor potential pathogens remains understudied, zoonotic transmission of pathogens such as *Salmonella* [30], Campylobacter [31], and West-Nile virus [32] from birds to humans have resulted in epidemics. Among the pathogens harbored by birds, the genus *Salmonella* is important because there are only two species of *Salmonella*, both of which have pathogenic strains that can result in Salmonellosis in humans [33]. Previous studies have detected *Salmonella* in raptors, including falcons, most likely as a consequence of eating infected prey [34]. Salmonellosis can also occur in captivity from infected feeds [35]. Understanding how *Salmonella* might affect the falcon gut microbiota is important in understanding its impact on falcon health and the increasing interaction between falcons and humans warrants a careful assessment of the pathogenic potential of the falcon gut microbiota. This will help in developing guidelines and best practices to safeguard both the health of falcons and individuals who come in contact with these birds.

Here, we characterize the gut microbiota of 29 falcons using amplicon sequencing of the V4 region of the 16S rRNA gene. We integrate this dataset with a previously published dataset consisting of 636 birds [16] that includes representative species from each of the 9 clades spanning the entire avian phylogeny [25]. Comparative analysis of the falcon gut microbiota in the context of avian evolution allowed us to test our hypothesis that a specialized uber-carnivorous diet plays an important role in shaping the falcon gut microbiota. Furthermore, we evaluate the contributions of *Salmonella*, a pathogenic bacterial genus, on the predicted functional pathways of the falcon gut microbiota. We hypothesized that the presence of *Salmonella* alters predicted bacterial pathways in the falcon gut.

## Materials and methods

### Ethics statement

This study was approved under an exempt protocol by the New York University IACUC Board. This observational study was carried out with consent from boarding facilities at four veterinary clinics in the United Arab Emirates (Royal Shaheen Dubai, Al Sayad Falcons Abu Dhabi, SNC Falcons Abu Dhabi and Al Dhafra Abu Dhabi) who voluntarily donated fecal samples collected using non-invasive procedures.

### Sample and data collection

Freshly produced fecal droppings were collected with sterile inoculation loops from 29 captive *Falco* birds and transported on ice-packs or dry ice to the laboratory within 4 hours of collection where they were stored at -80˚C until DNA extraction. Sampled species consisted of 11 purebred birds including *Falco cherrug* (n = 2), *Falco pelegrinoides* (n = 1), *Falco peregrinus* (n = 3), *Falco rusticolus* (n = 5), and 18 hybrid birds. Pure-bred and hybrid status was obtained from veterinary clinic records. A detailed description of the diet, flying activity, sex, and age of the birds in this study is provided in S1 Table.

### DNA extraction

Total DNA was extracted from the fecal samples using the Xpedition™ Soil/Fecal DNA Mini-Prep kit (Zymo Research, Irvine, CA, USA) following the manufacturer's protocol using roughly 0.25 g of fecal matter. Feces from the 29 birds were extracted in 1 to 4 replicates such that the total number of DNA samples was 42. DNA was eluted to a final volume of 100 μL in the elution buffer included in the extraction kit. Extracted DNA was quantified on the Nanodrop 8000 (ThermoFisher Scientific, Waltham, MA, USA) and stored at -80˚C until sequencing.

### PCR and 16S rRNA gene amplicon sequencing

The V4 hypervariable region of the 16S rRNA gene was amplified using the 515F-806R primer combinations [36] with the FastStart High Fidelity PCR system (Roche Applied Science, Penzberg, Germany). Agarose gel electrophoresis (2% gel, 90 V, 400 mA, and 30 minutes running time) was performed to confirm successful amplification of DNA. PCR products were purified using AMPure XP beads, DNA concentrations of the purified products were measured using a Qubit (ThermoFisher Scientific) and molarity was calculated based on the size of DNA amplicons as determined by a 2100 Bioanalyzer instrument (Agilent Technologies, Santa Clara, CA, USA). The average amplicon size was 411 bp and the average concentration was 12.9 ng/μL (SD: +/-11.5). Samples were diluted to 2 nM based on library size as recommended in the Illumina 16S Sample Preparation Guide, pooled at equimolar quantities, indexed, and multiplexed. Sequencing was performed on the MiSeq platform using the Reagent Kit v3 (Illumina, San Diego, CA, USA) to generate 250bp reads.

### Computational analysis

A total of 6,058,432 single-end raw reads were obtained from the 41 falcon samples (one sample was dropped due to low read count). In order to analyze these reads, we developed an analyses workflow that was first tested using a previously published dataset by Song et al. (EBI accession number PRJEB35449) [16] that consisted of 16S rRNA gene V4 region forward reads generated using the same primers as our dataset (515F-806R) from 2,135 vertebrates, including 1,074 birds, 747 mammals, and 314 other animals.

In order to infer the spontaneous errors introduced during sequencing, the newly sequenced falcon sequencing reads form this study and Song et al. datasets were initially processed individually for each sequencing run with DADA2 [37]. This allowed us to learn and adjust for the error rates in each sequencing run and infer sequence variants with high accuracy. After the error rate was inferred, all subsequent steps were performed jointly on both of these datasets using the phyloseq package [38] in R [4.2.0]. The reads were trimmed to 100 bp in order to retain high quality sequences (phred > 30). Reads with N nucleotides and/or > 2 expected errors were discarded (maxN = 0, maxEE = 2, truncQ = 2). The combined sequence table for the falcon and Song et al. datasets resulted in 272,103,907 reads. After chimera removal 261,167,457 (96%) reads were retained. Taxonomy was assigned using the RDP classifier [39] utilizing the SILVA v132 training set [40] as the reference. The R package DECIPHER [41] was used for multiple sequence alignment and a maximum likelihood tree was constructed with phangorn [42] using the neighbor-joining method.

A total of 255,109,025 reads (98%) belonged to the Song et al. [16] study, which consisted of 2 399 samples, including replicates. After retaining the replicates with highest coverage, a total of 229,672,486 reads belonging to 289,811 ASVs from 2,135 animals were initially retained. After removing ASVs with an abundance of less than 2 and presence in less than 5 samples, 187,673,522 reads and 24,267 ASVs remained from the 2,135 samples. Furthermore, non-bacterial reads (eukaryotes and fungi) and samples with sequencing depth of less than 10,000 reads were removed. These filtering steps resulted in a total of 187,014,663 reads (81%) and 24,267 ASVs from 1,824 animals. To prevent imbalances in species sample numbers, a maximum of 5 individuals per species were randomly selected and kept which resulted in a "comprehensive vertebrate dataset" (CVD) consisting of 136,976,389 reads and 24,065 ASVs from 1,330 animals from the Song et al. study, which was used to evaluate our workflow as described in S1 Fig.

Next, all non-avian samples were removed to create an "avian reference dataset" (ARD) which consisted of 61,469,984 reads and 10,603 ASVs from 665 birds (including our 29 falcons) spanning 9 phylogenetic clades (S1 File) [25]. This dataset was used to assess the falcon microbiota in the context of broader avian evolution.

Finally, the falcon microbiota data was analyzed independently. This dataset consisted of a total of 6,058,432 reads belonging to 843 ASVs from 41 falcon samples (29 falcons and 12 replicate samples). After filtering with the parameters described above, 5,618,239 reads and 145 ASVs were retained. To determine if there were differences between replicates, alpha and beta diversity analyses were performed. Statistically significant differences were not observed between replicates (S3G and S3H Fig); thus, the samples with the highest coverage were retained resulting in an "falcon specific dataset" (FSD) consisting of 4,368,722 reads and 109 ASVs from 29 falcons.

## Gut microbial diversity analyses

Alpha diversity was measured using species richness and Shannon's Diversity Index at the ASV level, calculated by rarefying reads to various depths between 1,000 and 10,000. One hundred iterations were performed at each depth and the mean values were used as the estimate of these measures in each sample. A maximum depth of 10,000 reads was chosen to include all individuals in the datasets (CVD, ARD and FSD). Kruskal–Wallis tests were used to assess the significance of differences in each of the alpha diversity metrics at the maximum depth (10,000 reads), by which the diversity had plateaued. Dunn's post-hoc test was performed to assess pairwise differences between groups. Generalized linear mixed effect models were used to evaluate associations between both measures of alpha diversity (response variables) and metadata

factors. For the ARD, phylogenetic clade, diet, flight status, GI tract region sampled, collection methods and captivity were used as explanatory variables. For the FSD four variables were available, namely age, sex, sampling location, and purebred status and all four were used as explanatory variables. The explanatory variables were treated to have fixed effects and random effects were assigned to each individual. P-value < 0.05 was considered statistically significant.

## Gut microbial composition analyses

Beta diversity was assessed at the genus level by log+1 transformation of the non-rarefied 16S rRNA gene count data for each sample followed by computing the unweighted and weighted UniFrac as well as the Bray–Curtis distances [38]. Principal Coordinate Analyses (PcoA) were performed using the phyloseq package [38] and visualized with the *ggplot2* package in R [43]. PERMANOVA and beta dispersion analysis was performed using the *vegan* package [44] using 10,000 randomizations where P-value < 0.05 was considered statistically significant.

## Clustering

Partitioning around medoids (PAM) clustering was performed on the ARD using the *cluster* package [45]. Individuals were grouped into multiple clusters (K = 2 to 14) based on the top seven principal coordinate axes obtained from the weighted UniFrac distances. Goodness of clustering was assessed using a "gap" statistic with 1,000 bootstrap replicates.

## Predictive functional abundances

PICRUSt2 was used to predict the functional contents of the falcon gut microbiota using the non-rarefied ASV counts [46]. A PCoA was performed for the 29 falcon samples with a log+1 transformation using Bray-Curtis distances with predicted MetaCyc [47] pathway abundances as features.

## Differential abundances of genera and function

Differential abundance of bacterial genera between dietary groups and predicted MetaCyc pathways between falcons with high, low, and not detected *Salmonella* load (*Salmonella* reads >100, 6–68, and 0 respectively; these cut offs were inferred from the variance in the absolute and relative abundances of *Salmonella* in the samples, S1 Table) was assessed with non-rarefied 16S rRNA gene abundance data using a negative binomial generalized linear model using the *differential expression analysis for sequence count data version 2* (*DESeq2*) package [48]. Only samples with reads > 10,000 were retained for the analyses mentioned above. Genera and predicted MetaCyc pathways with absolute $\log_2$[fold change] > 2 and adjusted p < 0.01 were considered significant. Multiple testing corrections were performed by computing FDRs using the Benjamini–Hochberg method.

## Random forests

All random forest classifiers were constructed using five-fold cross validation repeated three times with 500 trees using genera as predictors. The data was partitioned into training and validation sets as described below. The R-package *randomForest* [49] was used to build the trees and accuracy was used to select the optimal model. To train the classifier for phylogeny inferred clusters from the ARD dataset, 20 individuals per cluster were used for training and the remaining were used for testing. To train the classifier for dietary groups in the ARD (flesh-eaters, herbivores and piscivores) 10 individuals were used for training and 5 for testing. To train the phylogeny-diet classifier in the ARD, 11 individuals per group were used for

training and the remaining were used for testing. We assessed the performance of the classifiers by generating area under the receiver operating characteristic curves (AUC) using the R-package *ROCR* [50] and the default *varImp* function in the *caret* package [51] was used to calculate the variable importance.

## Results

### Sample description

In the United Arab Emirates, falcons are popular pets. They are housed in designated boarding facilities where they are fed a diet of freshly killed mice and birds including quail, pigeon, and chicken. We sampled freshly produced feces from 29 falcons housed in boarding facilities at four veterinary clinics in the United Arab Emirates. A detailed description of the diet, flying activity, sex, and age of the birds in this study is provided in S1 Table. All falcons included in this study were healthy.

### Avian reference dataset and factors associated with bird microbiota

In order to analyze the falcon fecal samples, we optimized the 16S analysis workflow presented by Callahan et al., [52] (S2 File) by reanalyzing a previously published dataset by Song et al. [16]. This dataset contained 2,135 animals sampled using methods consistent with this study and sequencing reads from the V4 region of the 16S rRNA gene, generated using the same primer set used in this study. After quality control (see Methods, S1A–S1F Fig), removal of low coverage samples (reads < 10,000) and balancing at the species level of the animals ($n \leq 5$), we generated a "comprehensive vertebrate dataset" (CVD, S1 File) consisting of 1,330 vertebrates, including 636 birds, 449 mammals, and 115 reptiles. Analyses of this dataset using our workflow reproduced several findings from the original manuscript (S1G–S1L Fig and S2 Table).

Next, we performed careful and detailed analysis of the CVD dataset to identify the factors contributing to gut microbial variation in birds, which was not performed in the original study [16]. To do so, we created an "avian reference dataset" (ARD) by retaining 636 birds with 5 representatives per species from each of the 9 avian clades [25]. This dataset is underrepresented in Falconiformes, although it consists of a small number of raptors including eagles and vultures (N = 15). A Principal Coordinate Analysis of the ARD using weighted UniFrac distances revealed that the avian gut microbiota composition is significantly associated with biological factors such as bird phylogeny (clade), captivity, and flight status ($p = 0.001, 0.001, 0.004$ respectively, *PERMANOVA*). Technical factors such as the gastrointestinal tract region sampled and sample collection methods also contributed appreciably, albeit with small effect sizes ($p = 0.001$ for both, *PERMANOVA*). A multivariable analysis that consisted of biological and technical variables revealed that the first Principal Coordinate axis (PCo1) was strongly associated only with phylogeny ($p < 2.2e-16$, *Generalized linear model*) (Fig 1A). On the other hand, PCo2 was associated with technical factors such as the gastrointestinal tract region sampled (feces vs intestine) and sample collection methods (ethanol vs freezing or RNALater) ($p = 0.002, < 2.2e-16$, and $0.004$, respectively, *Generalized linear model*). These observations were consistent when using unweighted UniFrac and Bray-Curtis distances as well as when samples across the avian clades were balanced (maxN = 50 per clade) to account for the dominance of Australaves in the data (Fig 1B). Moreover, birds could be clustered into five groups based on the microbiota data alone (Fig 1B and 1C). Furthermore, a random forest classifier was able to differentiate these five clusters with high accuracy (out of bag error = 19%, area under curve = 1, 1, 0.97, 0.99, and 0.95 respectively, S2A Fig). Alpha diversity was also associated with phylogeny ($p = 0.001$, *Kruskal-Wallis test*) with Palaeognathae having the highest

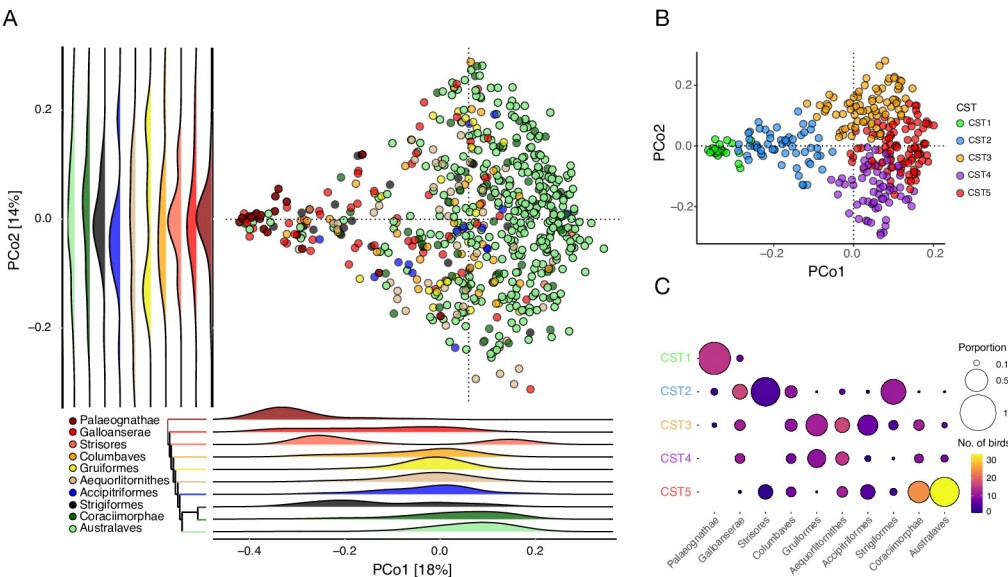

**Fig 1. Gut microbiota variation across bird phylogeny. (A)** PCoA using weighted UniFrac distances of the 636 birds in the Avian Reference Dataset colored by phylogenetic clades with each dot representing a bird. Legend shows color of each clade and the dendrogram is a reconstruction of the avian phylogenetic tree from [25]. Palaeognathae (maroon), a clade consisting of old-world birds such as ostriches, rheas and kiwis cluster on the left and Australaves (light-green), a clade consisting of recently evolved passerines and parrots, comprised opposite ends of the PCo1 spectrum with the rest of the clades occupying intermediary positions. Ridge plots show positions of phylogenetic clades along each PCo axis. **(B)** PCoA using weighted UniFrac distances on balanced Avian Reference Dataset (maxN = 50 per clade) retains global structure of gut microbiota variation in birds. Each dot represents a bird and colors represent their respective clusters (CST). Five major clusters were obtained using PAM clustering. **(C)** Balloon plot showing the phylogenetic distribution of birds among the five clusters inferred from (B). Consistent with the PCoA, the Paleognathae (old-world birds) and Australaves (parrots and passerines) aggregated into distinct clusters (CST 1 and 5 respectively) while the remainder of the birds were spread across the five clusters. Colors and size of the circles represent the number of birds per clade and their proportions, respectively.

alpha diversity and Australaves having the lowest among the 9 clades (S2B Fig). These analyses collectively indicate that host evolutionary histories play an important role in shaping the avian gut microbiota.

Interestingly, neither the top PCo axes nor the clusters were clearly associated with diet, potentially reflecting the interspecies heterogeneity in diet and underrepresentation of flesh-eating carnivorous birds in this dataset (N = 15). Thus, we hypothesized that a balanced dataset consisting of equal numbers of birds with distinct dietary habits may reveal the effect of diet on the bird gut microbiota. To evaluate this hypothesis, we identified a subset of birds from the intermediate clades (accounting for phylogeny as a confounding variable as we see Paleognathae and Australaves have distinct microbial composition in Fig 1A) whose diets were dominated by flesh, plants, and aquatic organisms. Flesh-eaters included uber-carnivorous raptors such as vultures and eagles (N = 15), herbivores included hummingbirds and pigeons whose diet is dominated by plants and grains (N = 15), and piscivores included penguins, flamingos and cranes that eat fish and crustaceans (N = 15). PCoA using the weighted UniFrac distance on this balanced dataset revealed clustering by diet ($R^2$ = 0.106, $p$ = 0.001, *PERMANOVA*, Fig 2A). Microbiota dispersion did not vary in the dietary groups ($p > 0.05$, *Betadisper*). Moreover, a random forest classifier was able to assign the flesh-eating, herbivorous, and piscivorous birds to their respective source dietary groups with 80%, 100%, and 80% accuracies respectively (OOB error = 20% and AUCs = 0.96, 0.96 and 1, S2C Fig).

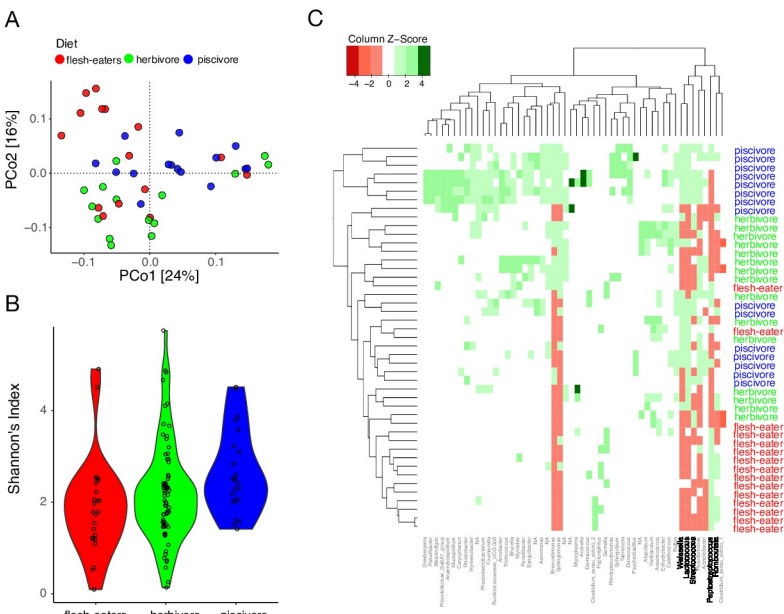

**Fig 2. Bird gut microbiota variation by diet. (A)** PCoA using weighted UniFrac differentiates the gut microbiota of birds with flesh-eating diet (red), mainly vegetarian diet (green), and mainly fish-eating diet (blue). Each dot is an individual bird. **(B)** Shannon diversity is highest in piscivores (*p* = 0.01, 0.03 respectively) but not significantly different between flesh-eaters and herbivores (*p* = 0.1). **(C)** Heatmap showing the relative abundance of the 52 taxa that are differentially abundant between the three dietary groups.

Furthermore, alpha diversity also varied significantly between the three dietary groups (*p* = 0.002, *Kruskal-Wallis test*) (Fig 2B). Piscivores had the highest Shannon's diversity but no significant differences were observed between the flesh-eaters and herbivores (*p* = 0.327, *Dunn's post hoc test*). A total of 52 genera were differentially abundant in the flesh-eaters compared to the other two dietary groups (log$_2$[fold change] > 2, *p* < 0.01, *DESeq*, S3 Table, S2D Fig). Flesh-eaters were enriched for *Peptostreptococcus* and *Romboutsia*, bacterial genera associated with high protein diets in animals [53, 54] and depleted in bacteria that assist in fiber digestion in the gastrointestinal tract namely, *Lactococcus*, *Streptococcus* and *Weissella* [55]. Both *Peptostreptococcus* and *Romboutsia* were largely depleted in herbivores and piscivores (Fig 2C).

## The falcon microbiota resembles that of raptors

To assess the falcon microbiota in the context of bird evolutionary history, we jointly processed the 16S rRNA gene V4 region sequences generated from 41 samples sequenced from 29 falcons in this study with the same parameters used to generate the datasets analyzed above (described in Methods) (S3A–S3F Fig). The replicate samples did not vary significantly from one another (*p* = 0.575, *PERMANOVA*, S3G and S3H Fig). After removing the replicates by discarding the samples with lower coverages (see Methods), 4,368,722 reads from 29 falcons remained in the ARD. Principal coordinate analysis of this dataset using the weighted UniFrac distance reproduced the overall pattern observed in Figs 1 and 3. The falcons from this study overlapped with the reference falcons in the ARD and they grouped with raptors (Fig 3A). Accipitriformes was the only clade, apart from the lowly sampled clade Strisores (n = 6), that was not significantly different from the falcons on both PCo axes (*p* > 0.05, *Dunn's post hoc test*). This indicates that diet may have a significant effect on the falcon gut microbiota.

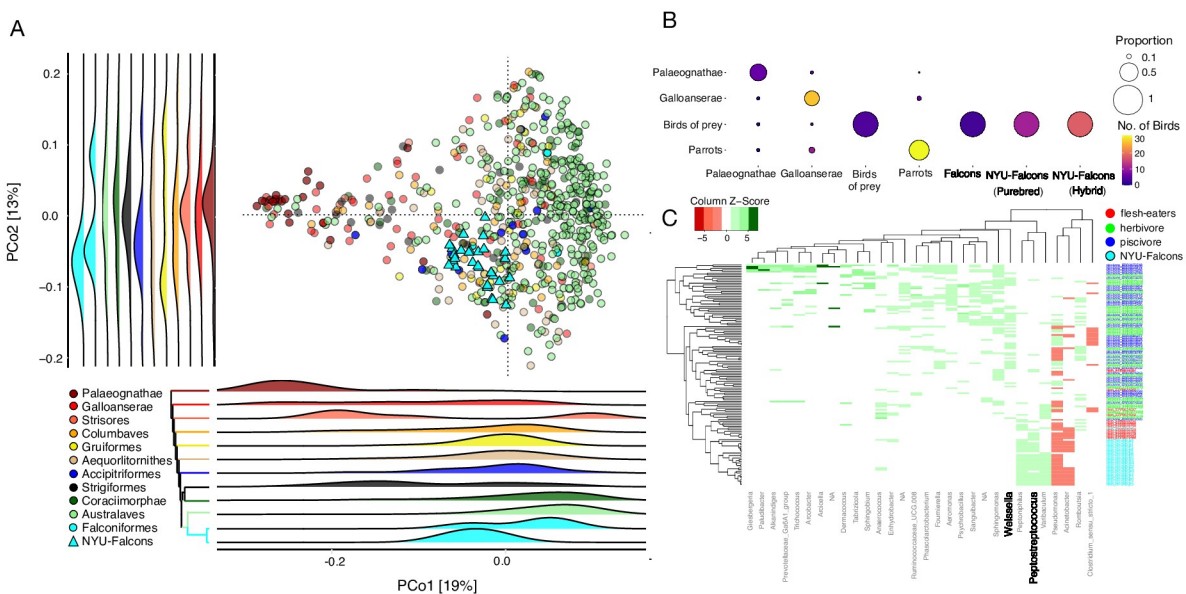

**Fig 3. Falcon gut microbiota resemble those of raptors. (A)** PCoA using weighted UniFrac distances including the 29 falcons from this study with the 636 birds from the Avian Reference Dataset. Each dot represents a bird and dots are colored by their respective phylogenetic clades. Our falcons are shown as cyan triangles while falcons from the Song et al. dataset are shown as cyan circles. Ridge plots show the positions of phylogenetic clades on each PCo axis. On PCo1 and PCo2 falcons and Accipitriformes the order that contains all raptors (blue), overlap with each other. **(B)** Balloon plot showing the number (color) and proportion (size) of birds classified to their respective phylogenetic clade by a random forest classifier using bacterial genera as features. All of the falcons, regardless of their source datasets and their purebred status get classified as raptors. Not a single falcon is classified as a parrot, genetically closest relative and vice-versa. **(C)** A heatmap consisting of 30 differentially abundant bacterial genera that differentiates the flesh-eating birds of prey, including the falcons from herbivores and piscivores.

To further investigate the effect of diet on the falcon microbiota, we used the random forest model trained to classify birds into their dietary groups (S2C Fig) and used 29 falcons from this study as the test dataset. All of the falcons from this study were classified as flesh-eaters. When parrots were used as the test dataset, 54% were grouped with herbivores and 44% with piscivores. We created a third random forest classifier model, this time encompassing both the phylogenetic and dietary groups to test if the falcon microbiota resemble those of their phylogenetic relatives or those of raptors with whom they share dietary preferences. This model included 44 birds from the ARD (excluding falcons from this study) evenly distributed across four categories (N = 11), three of which correspond to the clustering analysis above (Fig 1B): (i) old-world birds- Palaeognathae -representing Cluster 1, (ii) Galloanserae representing Clusters 2–4, (iii) parrots representing Australaves in Cluster 5 as well as the herbivores. For the fourth category, we included the raptors (Accipitriformes), which share a similar diet with falcons. If the falcon microbiota had stronger resonance with its phylogenetic cousins, they would be expected to cluster with parrots. If diet trumps phylogeny, they would cluster with the raptors. If their gut microbiota did not resemble either of these two groups, they would be expected to cluster with Galloanserae. The random forest model accurately classified the birds from the four source groups in the training dataset (OOB error = 11%). In the testing dataset, the model accurately classified 100% of the raptors (N = 4), 67% of the Palaeognathae (N = 9), and 79% of the parrots (N = 39) to their respective groups (Fig 3B). As expected, classification error was high for the Galloanserae because their gut microbiota does not cluster into a single group but spans multiple clusters (Fig 1C). All of the falcons in our dataset (N = 29) as well as the reference falcons included in the ARD (N = 3) were classified as raptors (Fig 3B).

Comparison of the 29 falcons in this study with the herbivores and piscivores included in the ARD revealed differential abundance of 85 genera ($\log_2$[fold change] > 2, $p < 0.01$, *DESeq*). Out of these 85 taxa, 30 genera also discriminated against the three diet groups in the previous analyses including birds from the ARD (Fig 2C). Hierarchical clustering analysis using the relative abundance of these 30 genera revealed that the 29 falcons in this study cluster with flesh-eating raptors and not with the herbivores and piscivores in the ARD (Fig 3C). Both groups of flesh-eating birds are similarly enriched in *Peptostreptococcus* and depleted in *Weissella*. Neither of these genera were significantly different between the falcons in this study and raptors in the ARD ($p = 0.09$ and $0.47$, *Kruskal Wallis test*). These analyses collectively indicate that the falcon gut microbiota resembles that of the raptors rather than that of their phylogenetically closest relatives—parrots—potentially owing to a large effect of carnivorous diet.

## *Salmonella* is associated with the falcon gut microbiota functions

We analyzed the gut microbiota of the falcons from this study independently of the avian reference dataset (Fig 4). Firmicutes was the dominant phylum (73% of total reads) in the falcon gut followed by Actinobacteria and Proteobacteria (Fig 4A). These observations are consistent with previous studies that have found falcon gut microbiomes to consist of *Firmicutes*, *Proteobacteria* and *Actinobacteria* [17–20, 22–24]. All but one of the studies have small sample sizes (n = 1–11, S4 Table) [18–20, 22–24]. Gut microbiota composition did not vary significantly by sex, age or sampling site and no differences were observed between the purebred and hybrid birds sampled in this study ($p > 0.05$, PERMANOVA) (S3I Fig). However, PCo1 obtained

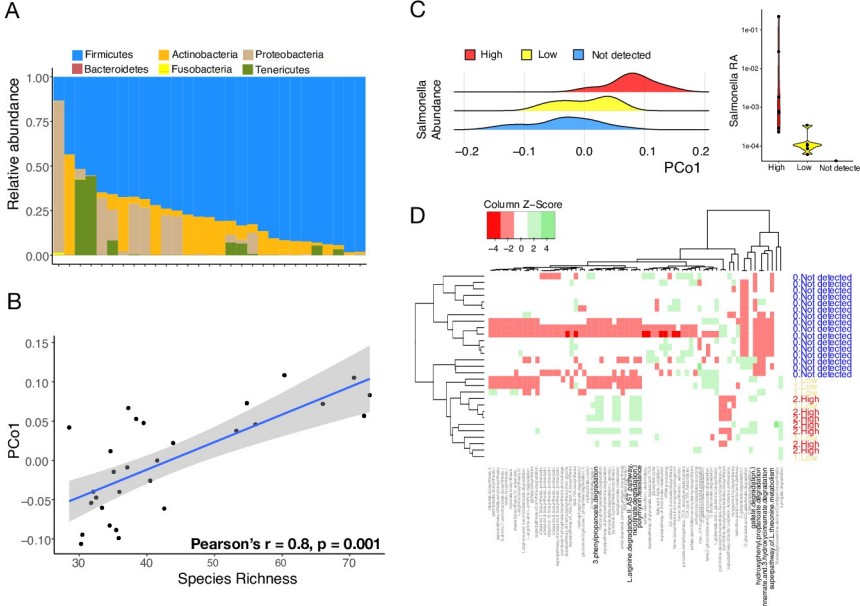

**Fig 4. Falcon gut is dominated by Firmicutes and predicted functional pathways associated with *Salmonella*. (A)** Abundance of gut bacterial phyla in the 29 falcons from this study. Firmicutes is the most dominant phyla followed by Actinobacteria, Proteobacteria and Tenericutes. **(B)** Pco1 was strongly correlated with species richness (Pearson's r = 0.8, $p = 0.001$) in falcons from this study. **(C)** PCoA analysis using Bray-Curtis distances of PICRUSt2-generated MetaCyc pathway abundances reveals difference in falcon gut microbiota functions by *Salmonella* abundance (left). Boxplot showing relative abundance of *Salmonella* (log scaled) across the three groups (right). **(D)** Heatmap of 69 differentially abundant pathways differentiating the birds with and without *Salmonella*. Dendrogram from hierarchical clustering (left) separates individuals with undetected *Salmonella*. Differentially abundant pathways associated with immunity and pathogenicity are shown in bold.

using weighted UniFrac distance was positively associated with species richness (Pearson's r = 0.8, *p* = 0.001, *Generalized linear model*, Fig 4B).

We found *Salmonella* to be the 11[th] most abundant genus in the falcons sampled in this study. All species of *Salmonella* are considered harmful pathogens for humans and they are naturally abundant in raptors [56]. We assessed whether presence of *Salmonella* is associated with gut microbial diversity, composition, and functional potential of the falcon gut microbiota. Relative abundance of *Salmonella* was neither associated with the gut microbial diversity (*p* > 0.05, *Kruskal-Wallis test*) nor composition assessed using the first 3 PCo axes obtained using weighted UniFrac distances (*p* > 0.05, *Kruskal-Wallis test*). Next, we evaluated gut microbial functional potential using PICRUSt2 [46] to predict abundances of MetaCyc pathways. A Principal Coordinate Analysis using Bray-Curtis distance with the predicted pathways as features revealed the falcon gut functional composition is strongly associated with *Salmonella* load ($R^2$ = 0.2, *p* = 0.01, *PERMANOVA*) and not with age, sex, sampling site and hybrid status (*p* > 0.05, *PERMANOVA*). A clear differentiation along the Pco1 was observed between the falcons with high, low, and not detected *Salmonella* load (*p* = 0.003, *Kruskal-Wallis test*, Fig 4C). Furthermore, a total of 69 predicted pathways were differentially abundant between the falcons with not detected *Salmonella* compared to high or low *Salmonella* loads (Fig 4D, S5 Table). Only a few of these pathways are specific to the *Salmonella* genome (n = 3). Several of these differentially abundant pathways are associated with protein buildup and maintenance, immune regulation, and breakdown of antimicrobial compounds, indicating that *Salmonella* presence may potentially alter the functional capacity of the falcon gut microbiota (S6 Table).

## Discussion

The modern birds radiated 66 million years ago [57] and today, there are over 10,000 species of birds that have adapted to diverse ecological niches worldwide resulting in tremendous morphological, physiological, and dietary specializations [58]. Since the gut microbiota may play a critical role in regulating vertebrate physiology, it is important to understand its role in avian evolution and their adaptations across diverse habitats [12]. Although microbiota studies of non-domesticated avian species using 16S rRNA gene sequencing are emerging, each study consists of limited samples–potentially reflecting difficulties in collecting samples from non-commercial birds [59]–which may lead to inconsistent findings across studies. Importantly, it is difficult to combine data from different studies due to variations in experimental techniques such as sample collection methods, DNA extraction protocols, and the choice of 16S rRNA gene region for amplification [12, 60], which prevent meaningful comparisons between studies. Therefore, it is imperative that future avian studies minimize variations in technical factors to make multi-study comparisons possible. A recent study [16] ameliorated some of these challenges by analyzing amplicons of the 16S rRNA gene V4 region using the universal primers (515F-806R) from 1,074 birds across the avian phylogeny along with hundreds of other animals, although raptors were severely underrepresented. In this study, we processed and reanalyzed this dataset and integrated much needed novel gut microbiota data from 29 falcons to create a comprehensive "avian reference dataset" consisting of 665 birds spanning 9 phylogenetic clades. The analysis of the avian reference dataset using our workflow reproduced previous results. For example, avian gut microbiota was significantly associated with physiological features such as flight status and captivity [16, 61]. Therefore, this dataset and the workflow may serve as an important resource for the avian microbiota community.

In addition to corroborating previous findings, we performed a more careful and detailed analyses on the publicly available dataset, which resulted in novel insights. Our results revealed a robust link between the avian evolutionary history and the gut microbiota. Multiple analyses

revealed strong differences in both gut bacterial diversity and composition between the two most phylogenetically diverged bird groups–Paleognathae (old-world birds) and Australaves (recently evolved passerines and parrots), indicating that avian gut microbiota is shaped partly by the host evolutionary history. This finding is consistent with previous observations that the degree of similarity between microbiota resembles evolutionary histories in birds [61, 62]. The lack of gut microbial differences between the intermediate clades could be a potential consequence of the limited taxonomic resolution provided by the 16S rRNA gene sequencing reads. Future studies incorporating shotgun whole metagenomics sequencing may reveal additional bacterial strains and their functions relevant in avian evolution.

Our results also demonstrate that diet is significantly associated with the bird gut microbiota. The effect of diet on gut microbiota was not detected when 636 birds in the avian reference dataset were considered together, likely due to the dietary heterogeneity within each clade [63]. However, analysis of a selected subset of birds whose diets are dominated by flesh, plant, or aquatic organisms revealed that diet has a significant effect on bird gut bacteria. Of the 52 genera differentially abundant between the three dietary groups, bacterial genera such as *Peptostreptococcus* and *Romboutsia* that are associated with high protein diets in vertebrates [53, 54] were also enriched in raptors whose diet are dominated by animal flesh. *Peptostreptococcus* is a genus of bacteria that is part of the normal gut flora of vultures [54, 64]. Previous studies have found elevated levels of lactic acid bacteria such as *Lactococcus*, *Streptococcus* and *Weissella* in herbivorous birds compared to flesh eating birds [62]. Presence of such bacteria helps metabolism of plant-based fibers via fermentation in the intestine [55, 65, 66]. These bacteria were significantly depleted in the falcons in this study as well as in raptors generally, indicating that diet plays an important role in shaping the bird microbiota. We note that 16S rRNA data is limited in its resolution and shotgun metagenomics would be better able to elucidate the associations between diet and key discriminating taxa.

Among the birds, the raptors are unique as their diet comprises almost exclusively of animal flesh. However, studies investigating their gut microbiota are scarce. As a result, whether genetics, evolutionary histories, and diet have a measurable effect on the raptor gut microbiota has remained unclear [17–24]. We sequenced falcons of multiple species in this study and found that the gut microbiota composition of the falcons did not differ between sex, age, or purebred status, indicating that genetic variations in the falcon genome may have minor contributions in shaping their gut microbiota. By combining the falcon gut microbiota sequenced in this study with the avian reference dataset, we were also able to evaluate the potential effect of evolutionary histories and diet on the falcon gut. Our results show that the gut bacteria of falcons are more similar to those of raptors rather than phylogenetically adjacent parrots, indicating that diet has played a prominent role in shaping the falcon gut microbiota. This finding is consistent with previous observations that diet and flight associated adaptations are strong drivers of gut microbiota variation compared to host phylogeny in modern birds [61].

The falcon gut microbiota warrants additional attention because they are known to harbor enteric zoonotic pathogens such as avian influenza virus [67], West Nile virus [68], Newcastle disease virus [69] and *Aspergillus* molds [70, 71] that can cause deadly human diseases and can be transmitted to humans via the practice of falconry [26]. The 10 most abundant genera we detected in the falcon gut were *Clostridium sensu stricto 1*, *Paeniclostridium*, *Fructobacillus*, *Peptostreptococcus*, *Ureaplasma* and *Peptoniphilus* along with some taxa that were classified only up to the family level including *Enterobacteriaceae*, *Clostridiales_Family_XIII* and *Coriobacteriaceae* (S7 Table). These are commensal members of the gut microbiota in vertebrates [72–75], although some of the species belonging to these may be pathogenic to birds and humans (e.g. *Clostridium sensu stricto 1*). Due to the limited taxonomic resolution of the 16S rRNA sequencing, we were unable to distinguish between the commensal and pathogenic

species in our data. Interestingly, we found appreciable levels of *Salmonella* in our falcons, which is consistent with previous reports [35]. Since *Salmonella* is a foodborne pathogen, infected meat is the most probable source. Although the falcons were fed a diet of freshly killed mice and birds including quail, pigeon, chicken in the boarding facilities, which may have harbored *Salmonella* [76]. Alternatively, they could have acquired it from the wild when the falcons were taken on hunting excursions. Future strain-level metagenomics sequencing followed by laboratory cultivation may enable the distinction between wild versus commercial *Salmonella* strains and their pathogenic potential.

All species of the genus *Salmonella* are zoonotic pathogens that can cause deadly foodborne illnesses in humans but infected birds may remain asymptomatic [77]. Despite appreciable *Salmonella* load, the falcons in our study did not show any apparent signs of illnesses and were not being treated for any infections. The lack of disease symptoms poses a significant risk to humans that come in direct or indirect contact with these birds (falcon handlers, owners, facility cleaners, etc.). Because of the widespread pathogenicity across this genus, species level resolution was not required for us to evaluate its potential impact on the falcon gut microbiota. The relative abundance of *Salmonella* was strongly associated with shifts in predicted functional potential of the falcon gut (S5 Table). *Salmonella* high samples in our study showed elevated abundances of L-Arginine and L-Threonine degradation pathways. Both are amino acids that are essential for building protein mass, sustaining proper protein balance, and maintaining immune homeostasis in animals [78, 79]. These findings indicate that the presence of *Salmonella* in the gut may negatively influence the overall health of the falcons by affecting the capacity to build and maintain protein mass, which is essential for flighted birds that engage in long-distance migrations. Moreover, small aromatic compounds such as gallic acid, 3-phenyl-propionic acid, nicotinic acid, 4-methylcatechol, 3-hydroxycinnamate and their derivatives are known antimicrobials [80–84] and are known to have *Salmonella* inhibitory effects in humans [85]. The falcons in our study that harbored high levels of *Salmonella* showed enrichment of functional pathways that degrade these antimicrobial compounds. Pathways associated with resistance of antimicrobial drugs such as polymyxin were also enriched in the falcons with high *Salmonella* load. These results indicate that *Salmonella* presence lowers the immune capacity of the falcons, making them susceptible to harboring infectious agents that can be transmitted to humans via falconry [86].

In addition to *Salmonella*, the falcons may have harbored additional pathogenic bacteria, which we were unable to detect in this study due to limited taxonomic resolution of the 16S rRNA gene sequencing. Since, this was a cross sectional study where the birds were sampled at a single timepoint, the duration of infection also remained unclear. Future studies involving longitudinal sampling may reveal the duration and recurrence of intestinal pathogens. Such studies should also consider implementing additional techniques (e.g. qPCR) for robust quantification of pathogenic bacterial load, which may determine the degree of gut microbial variation. Finally, the *Salmonella* associated differences in functional pathways we describe in this study are computational predictions based on 16S rRNA. Although these changes indicate gut microbiota may function differently in presence of an enteric pathogen, detailed characterization of functional differences in response to a particular pathogen will require strain level whole metagenomics sequencing followed by strain isolation and coculture experiments in the future. Moreover, a previous study reported that captivity may have a detectable effect on the falcon gut microbiota [17]. Future studies should consider including samples collected from the wild, when possible, to assess the overall effect of captivity. Despite these limitations, our results show that pathogenic bacteria leave a detectable signature in the gut microbiota of captive falcons. Therefore, pathogen profiling of the gut before reintroduction of captive birds to their natural habitats may help improve outcomes for conservation efforts.

## Supporting information

**S1 Fig. QC of Song et al. dataset and workflow validation.** Sequencing depth for each sample and each taxon **(A)** before filtering any reads, and **(B)** after removal of lowly abundant ASVs; this does not reduce sequencing depth of either the samples or the taxa significantly. **(C)** Abundance of Phyla in the dataset. Fungal and eukaryotic phyla were removed from the dataset. **(D)** After filtering, 35 phyla and a total of 24,000 taxa remained. Rarefaction curves for **(E)** species richness and **(F)** Shannon diversity show plateauing alpha diversity for most samples starting at a rarefaction depth of 10,000 reads. **(G)** Principal coordinate analysis using unweighted UniFrac distances at the ASV level and randomly sampled to include a maximum of 5 individuals per species (1,330 points). Non-flighted mammals (purple dots) have a microbiota distinct from birds (yellow circles) and bats (purple boxes) cluster with birds. **(H)** Violin plot of PCo1 shows bird and bat microbiota are not significantly different (p = 0.238, Dunn's post hoc test). **(I)** Shannon diversity and species richness for flighted and non-flighted birds and mammals. Shannon's diversity for all pairwise comparisons had p < 0.05, except for flighted mammals vs flighted and flightless birds (p = 0.470, 0.111). For Richness all pairwise comparisons had p < 0.05, except for flighted mammals vs flighted and flightless birds (p = 0.361, 0.161). **(J)** Shannon's diversity for migratory and non-migratory birds have significantly different alpha diversities (p = 0.017). Richness between migration modes is not significantly different (p = 0.088). **(K)** Alpha diversity (richness) scaled against body mass for mammals (R2 = 0.450, p < 2.2e-16) and birds ($R^2$ = 0.367, p < 2.2e-16). **(L)** In non-flighted mammals alpha diversity scales with body mass ($R^2$ = 0.433, p < 2.2e-16) but in flighted mammals (bats) the relationship is insignificant ($R^2$ = 0.0788, p = 0.808). In both flighted ($R^2$ = 0.343, p = 0.018) and non-flighted birds ($R^2$ = 0.334, p < 2.2e-16) alpha diversity scales with body mass albeit less than with non-flighted mammals.
(PDF)

**S2 Fig. Phylogeny and diet supplementary figs. (A)** A random forest classifier validates the grouping inferred from the clustering with 19% OOB error and AUCs = 1, 1, 0.97, 0.99 and 0.95. **(B)** Alpha diversity correlates with phylogeny, with Palaeograthae having the highest (p < 0.05 for all comparisons except with Galloanserae). **(C)** A random forest classifier supports the beta diversity analysis with 20% OOB error and AUCs = 0.96, 0.96 and 1 for the three dietary groups. **(D)** DESeq analysis reveals 52 bacterial taxa differently abundant between the dietary groups ($\log_2$[fold change] > 2, $p < 0.01$).
(PDF)

**S3 Fig. Application of workflow to falcon dataset and QC steps.** Sequencing depth for each sample and each taxon **(A)** before filtering any reads, and **(B)** After removal of lowly abundant ASVs; this does not reduce sequencing depth of either the samples or the taxa significantly. **(C)** Total abundance and prevalence of phyla in the dataset. **(D)** After filtering, 6 phyla and a total of 109 taxa remained. One sample was removed on account of low reads, with the resulting phyloseq object having 29 samples. Rarefaction curves for **(E)** species richness and **(F)** Shannon's diversity show plateauing alpha diversity for most samples starting at a rarefaction depth of 15,000 reads. **(G)** Beta diversity analysis of falcons with duplicates. No marked differences between replicates observed p > 0.05, PERMANOVA. Samples in blue are replicates with higher read counts and samples in red are replicates with lower read counts. **(H)** Alpha diversity between replicates does not differ markedly as well (p > 0.05 at rarefaction depth = 15,000). **(I)** PCoA using weighted UniFrac distances of the falcon 16S rRNA gene data reveals separation that is not explained by species (color) or sampling site (shape).
(PDF)

**S1 Table. Falcon metadata.**
(XLSX)

**S2 Table. PERMANOVA results on PCoA.** PERMANOVA results on weighted UniFrac ordination reveal strong correlation of microbiota with phylogeny in mammals and to a weaker extent in birds. Asterisks denote significance (P < 0.05).
(XLSX)

**S3 Table. Genera differentially expressed between dietary groups from DESeq analysis.**
(XLSX)

**S4 Table. Studies on falcon gut microbiota.**
(XLSX)

**S5 Table. MetaCyc pathways differentially abundant between *Salmonella* high, low and no samples.**
(XLSX)

**S6 Table. Subset of MetaCyc pathways that are differentially abundant between *Salmonella* high, low and no samples.**
(XLSX)

**S7 Table. Fifteen most abundant genera with individual ASVs listed and percentage of reads assigned to the genera out of total reads.**
(XLSX)

**S1 File. Phyloseq object for ARD.**
(ZIP)

**S2 File. Workflow for 16S analysis.**
(ZIP)

## Acknowledgments

We thank M. M. Dieng, R. Shraim and M. Vinu for their contributions to the laboratory work and bioinformatics expertise respectively. We thank the Center for Genomics and Systems Biology, NYUAD Core Bioinformatics and Technology Platforms for assistance with technical work. We thank Al Sayad Falcons, Al Dhafra Falcons, SNC Falcons, and Royal Shaheen Events for providing access to falcon samples and for their assistance with sampling. We thank S. Al Dhabari, P. Bergh and the falconry community for the hospitality and expertise.

## Author Contributions

**Conceptualization:** Anique R. Ahmad, Youssef Idaghdour, Aashish R. Jha.

**Data curation:** Anique R. Ahmad.

**Formal analysis:** Anique R. Ahmad, Samuel Ridgeway.

**Funding acquisition:** Youssef Idaghdour, Aashish R. Jha.

**Investigation:** Anique R. Ahmad, Ahmed A. Shibl, Aashish R. Jha.

**Methodology:** Anique R. Ahmad, Aashish R. Jha.

**Resources:** Youssef Idaghdour, Aashish R. Jha.

**Supervision:** Youssef Idaghdour, Aashish R. Jha.

**Visualization:** Anique R. Ahmad, Ahmed A. Shibl.

**Writing – original draft:** Anique R. Ahmad, Samuel Ridgeway, Ahmed A. Shibl, Aashish R. Jha.

**Writing – review & editing:** Anique R. Ahmad, Samuel Ridgeway, Ahmed A. Shibl, Youssef Idaghdour, Aashish R. Jha.

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
