## [Decision Letter · Decision Letter 0]

18 Aug 2023

PONE-D-23-18286Falcon gut microbiome is shaped by diet and enriched in SalmonellaPLOS ONE

Dear Dr. Jha,

Thank you for submitting your manuscript to PLOS ONE. After careful consideration, we feel that it has merit but does not fully meet PLOS ONE’s publication criteria as it currently stands. Therefore, we invite you to submit a revised version of the manuscript that addresses the points raised during the review process.

Both reviewers have major concerns but the idea of the manucript is very good. In particular a better structuring and focus of the manuscript is necessary- please be directed to the reviewers commtns. Additional missing literature should be added and the influence of food towards the microbiom of the birds should be discussed more detailed and should led to a more open critism of the own data.

We look forward to receiving your revised manuscript.

Kind regards,

Michael Lierz, Dr. med vet

Academic Editor

PLOS ONE

“We thank M. M. Dieng, R. Shraim and M. Vinu for their contributions to the laboratory work and bioinformatics expertise respectively. We thank the Center for Genomics and Systems Biology, NYUAD Core Bioinformatics and Technology Platforms for assistance with technical work. We thank Al Sayad Falcons, Al Dhafra Falcons, SNC Falcons, and Royal Shaheen Events for providing access to falcon samples and for their assistance with sampling. We thank S. Al Dhabari, P. Bergh and the falconry community for the hospitality and expertise. This work is funded by NYUAD Capstone funds to Samuel Ridgeway, NYUAD grants ADHPG AD318 to A. R Jha and grant AD105 to Y. Idaghdour.”

Additional Editor Comments:

Both reviewers have major concerns but the idea of the manucript is very good. In particular a better structuring and focus of the manuscript is necessary- please be directed to the reviewers commtns. Additional missing literature should be added and the influence of food towards the microbiom of the birds should be discussed more detailed and should led to a more open critism of the own data.

Reviewers' comments:

Reviewer's Responses to Questions

**Comments to the Author**

1. Is the manuscript technically sound, and do the data support the conclusions?

Reviewer #1: Partly

Reviewer #2: Partly

2. Has the statistical analysis been performed appropriately and rigorously? 

Reviewer #1: Yes

Reviewer #2: Yes

3. Have the authors made all data underlying the findings in their manuscript fully available?

Reviewer #1: Yes

Reviewer #2: Yes

4. Is the manuscript presented in an intelligible fashion and written in standard English?

Reviewer #1: Yes

Reviewer #2: No

5. Review Comments to the Author

Reviewer #1: The authors used a cultivation independent analysis to study the Falcon gut microbiota using 16S rRNA gene amplicon NGS sequencing. This methods gives a qualitative, non-quantitative inside into the microbial community diversity and composition of the falcon gut microbiota. Beside the generated dataset, public datasets were implemented for comparative analysis.

Overall the paper was written in a clear structure explaining the used methods in detail. At few points the analytic steps must be checked and data points and discussion points are recommended to be added.

I missed more details to the studied Falcons.

Literature is missing:

I checked pubmed just for “Falcon + Salmonella” there is literature available not considered here. The authors should implement topic specific literature

rDNA should not be used, should be changed to 16S rRNA gene; I know that the shortcut it is used by some scientists , however it is not correct

Line 16/17: it would be helpful here to point out if wild or captive or even captive-bred falcons are studied here, that makes huge differences and should be mentioned already in the abstract

Line 20: correct to relative abundance

Line 23: “consists of “ ? better “contains/ includes ….” ?

Line 55: dot after (25)

Lines 74: I think in the first step epidemics are not a key problem, individual people/ workers that are indirect contact to the Falcons must be aware of the problem.

Lines 83: remove “generated”

Line 89/90: what means “functional potential” should be specified.

I would prefer to formulate a hypothesis, what was expected from the gut microbiome of Falcons.

Lines 108: it must be specified if the birds came from the same or different locations.

Line 111: Total DNA instead of bacterial DNA

Line 114: 100 µl of what? How was DNA stored? Were the DNA measurements realistic measurements, meaning was a clear DNA peak visible at OD260 in the nanodrop? If not I would not give DNA values here. For environmental samples photometric values are a poor estimate.

Lines 182: level of alpha diversity analysis should be added, ASVs?

Line 199: I think a per sample standardization is usual done after transformation before in between sample distances are calculated ? - same see line 214?

Line 220-221: variance in relative abundance is more important. Which should also be used for grouping. Please check. Was this considered.

Line 243: first time it was mentioned that birds were in medical clinics.

Line413-415: Correlation of Beta with Alpha-diversity data is uncommon. Isn’t it a correlation of the same data set. Also Beta diversity measures consider species abundance. I would remove that analysis.

Line 430: it would be good to give the relative abundances here for high and low load that the reader gets directly an impression on the key abundance of Salmonella

It would be interning which genera were the other most abundant once, no discussion is given on them.

Line 442: no -> not detected

Lines 444-446: How well is this prediction (% similarity values available?)

Lines 442-448: I am generally careful for those predictions. How precise are those correlations.

Have the authors also checked for the abundance of other pathogens?

I would prefer a list of ASVs assigned to specific pathogens. Was really nothing detected?

It is clear that those data are just hinds, but never the less, interesting to give,

Line 477: “Stark” is that really English or German?- better “strong”?

Lines 536-540: however a list of taxa with potential pathogens can be given and discussed

I would consider more details in the discussion, especially to the limited quality of information obtained by the amplicon approach:

- When the Salmonella load is discussed I would check a consistent colonization. Each falcon was analyzed just once. Must not be done in this paper, but should be discussed.

- It was not discussed if the food/food source may be the reason for a Salmonella contamination. For readers as me which are not Falcon experts it would be interesting to get more information of the nutrition and food sources in the country of analysis. I would consider a clear link to food source. It the food quality controlled? Can that be the primary source for Salmonella infections…. All questions the paper opens to me. To get respective information in the paper will enhance the quality.

- Amplicon data are all just relative abundance data, I would recommend to perform a quantitative detection of the Salmonella load (infA qPCR works for example well). If DNA samples are still in a freezer that can be done quickly and can be added to the paper. Otherwise again it should be at least discussed as important control in subsequent studies.

- Cultivation of respective Salmonella should be recommended. This again would enhance the knowledge in the growth state of Salmonella, alive, active -> culturable, dead or VBNC? Again should be at least discussed. I think there are respective studies available which worked on that.

Reviewer #2: The manuscript by Ahmad and colleagues focuses on falcon microbiota. The aim of the manuscript is very interesting and fills the gap in knowledge about wild birds’ microbiota. However, the manuscript is confusing, it supposed to be focusing on falcons’ microbiota but spend a lot of time on reanalyzing somebody else data. The authors need to decide what they want to accomplish: provide comprehensive analysis of falcon microbiota or perform meta-analysis of birds’ microbiota. I think the former would be more valuable to scientific community.

Additional comments:

1. Microbiome refers to microbiota, their function, metabolome, etc. The paper only focusses on composition and predicted function; therefore, microbiota should be used instead microbiome.

2. Line 76-81: this should be included in Result section when the metanalysis is described or in discussion section.

3. Line 117, 260-280 ratio close to 1 indicates poor DNA quality. Have you performed any other quality check?

4. Line 119-130: have the library been indexed? What was the sequencing length?

5. Line 248: new 16S analysis workflow was created, why? Why not follow already established workflow for 16S analysis? What is the advantage of the new workflow?

6. PLOS authors have the option to publish the peer review history of their article (what does this mean?). If published, this will include your full peer review and any attached files.

Reviewer #1: No

Reviewer #2: No

---

## [Author Response · Author response to Decision Letter 0]

6 Oct 2023

Authors’ response to the Editor and Reviewers comments

Author’s response: The formatting requirements have been met.

“We thank M. M. Dieng, R. Shraim and M. Vinu for their contributions to the laboratory work and bioinformatics expertise respectively. We thank the Center for Genomics and Systems Biology, NYUAD Core Bioinformatics and Technology Platforms for assistance with technical work. We thank Al Sayad Falcons, Al Dhafra Falcons, SNC Falcons, and Royal Shaheen Events for providing access to falcon samples and for their assistance with sampling. We thank S. Al Dhabari, P. Bergh and the falconry community for the hospitality and expertise. This work is funded by NYUAD Capstone funds to Samuel Ridgeway, NYUAD grants ADHPG AD318 to A. R Jha and grant AD105 to Y. Idaghdour.”

Author’s response: The information in the acknowledgements section has been removed. We would like to update the Funding Statement as follows: 

 “This work is funded by NYUAD Capstone funds to Samuel Ridgeway, NYUAD grants ADHPG AD318 to A. R Jha and ADHPG AD105 to Y. Idaghdour.”

We have also added this statement to the cover letter.

Additional Editor Comments:

Both reviewers have major concerns but the idea of the manucript is very good. In particular a better structuring and focus of the manuscript is necessary- please be directed to the reviewers commtns. Additional missing literature should be added and the influence of food towards the microbiom of the birds should be discussed more detailed and should led to a more open critism of the own data.

Author’s response: We thank the Editor for considering our manuscript for publication in PLoS One. We have incorporated the valuable comments and feedback from the two reviewers and are submitting a revised manuscript for your consideration.

Reviewers' comments:

Reviewer's Responses to Questions

Comments to the Author

1. Is the manuscript technically sound, and do the data support the conclusions?

Reviewer #1: Partly

Reviewer #2: Partly

Author’s response: We have integrated the Reviewers’ comments and feedback and revised our manuscript, which has improved the clarity of our manuscript. We are grateful to the two reviewers for their suggestions.

2. Has the statistical analysis been performed appropriately and rigorously?

Reviewer #1: Yes

Reviewer #2: Yes

Author’s response: We thank the two reviewers for their support of this manuscript.

3. Have the authors made all data underlying the findings in their manuscript fully available?

Reviewer #1: Yes

Reviewer #2: Yes

Author’s response: We have made all data and analysis scripts fully available. We hope these will be useful resources to the scientific community.

4. Is the manuscript presented in an intelligible fashion and written in standard English?

Reviewer #1: Yes

Reviewer #2: No

Author’s response: We have corrected minor grammatical errors and typos. We have also restructured parts of the manuscripts for brevity and to increase clarity.

5. Review Comments to the Author

 

Reviewer #1: The authors used a cultivation independent analysis to study the Falcon gut microbiota using 16S rRNA gene amplicon NGS sequencing. This methods gives a qualitative, non-quantitative inside into the microbial community diversity and composition of the falcon gut microbiota. Beside the generated dataset, public datasets were implemented for comparative analysis.

Overall the paper was written in a clear structure explaining the used methods in detail. At few points the analytic steps must be checked and data points and discussion points are recommended to be added.

Author’s response: We thank the Reviewer for the detailed investigation of our manuscript. We are glad that the Reviewer found clarity in our manuscript. We have addressed the Reviewer’s specific comments below. Integrating these changes have significantly improved our manuscript.

I missed more details to the studied Falcons.

Author’s response: We have added a detailed description of the diet, flying activity, sex, and age of the birds in this study is provided in S1 Table.

Literature is missing:

I checked pubmed just for “Falcon + Salmonella” there is literature available not considered here. The authors should implement topic specific literature 

Author’s response: We have added key references on Salmonellosis in falcons in lines 73-75 of the revised manuscript.

rDNA should not be used, should be changed to 16S rRNA gene; I know that the shortcut it is used by some scientists , however it is not correct

Author’s response: We have changed rDNA to 16S rRNA throughout the revised manuscript.

Line 16/17: it would be helpful here to point out if wild or captive or even captive-bred falcons are studied here, that makes huge differences and should be mentioned already in the abstract

Author’s response: We have clarified that these are captive falcons in line 17 of the revised manuscript.

Line 20: correct to relative abundance

Author’s response: We have changed “abundance” to “relative abundance” in line 21 of the revised manuscript.

Line 23: “consists of “ ? better “contains/ includes ….” ?

Author’s response: We have replaced “consists of” with “contains” in line 24 of the revised manuscript

Line 55: dot after (25)

Author’s response: We have added a period in line 56 of the revised manuscript.

Lines 74: I think in the first step epidemics are not a key problem, individual people/ workers that are indirect contact to the Falcons must be aware of the problem.

Author’s response: The Reviewer raises a valid point. We have amended the sentence to bring attention to the risk that Salmonella possesses to individuals who come in contact with the birds. We are grateful to the reviewer for this suggestion. The revised sentences are in lines 78-79 of the revised manuscript

Lines 83: remove “generated”

Author’s response: Done.

Line 89/90: what means “functional potential” should be specified.

Author’s response: We have replaced “functional potential” with “predicted functional pathways” to improve clarity. We thank the reviewer for this point.

I would prefer to formulate a hypothesis, what was expected from the gut microbiome of Falcons.

Author’s response: We have added our hypotheses in lines 93-98 of the revised manuscript

Lines 108: it must be specified if the birds came from the same or different locations.

Author’s response: We have specified where the birds come from in lines 105-108 of the revised manuscript. We have also added a detailed description of each bird in S2 Table as specified in line 117.

Line 111: Total DNA instead of bacterial DNA

Author’s response: Changed in line 120.

Line 114: 100 µl of what? How was DNA stored? Were the DNA measurements realistic measurements, meaning was a clear DNA peak visible at OD260 in the nanodrop? If not I would not give DNA values here. For environmental samples photometric values are a poor estimate.

Author’s response: The DNA was eluted in 100 µL of the elution buffer included in the extraction kit and stored in -80°C until sequencing. We agree with the reviewer that nanodrop is not the best method to measure DNA concentration for environmental samples. Thus, we have removed the DNA concentrations values in the revised manuscript. These changes are in lines 123-125 of the revised manuscript.

Lines 182: level of alpha diversity analysis should be added, ASVs?

Author’s response: We have specified that the alpha diversity was calculated using ASVs in lines 193-194 of the revised manuscript.

Line 199: I think a per sample standardization is usual done after transformation before in between sample distances are calculated ? - same see line 214?

Author’s response: The Reviewer is correct that one way to standardize sequencing depth across samples is to standardize the number of reads per sample by rarefaction. However, McMurdie and Holmes (2014) argue against rarefaction in microbiome data analysis to avoid throwing away data from high coverage samples. Instead, the log+1 transformation is performed per sample which reduces variance in coverage across the dataset allowing for sample distances to be calculated. This approach was described in Callahan et al. (2016) and was implemented in this study. We have made modification to the sentence describing this point in lines 209-210 of the revised manuscript.

Line 220-221: variance in relative abundance is more important. Which should also be used for grouping. Please check. Was this considered.

Author’s response: The Reviewer raises a valid point. We did not consider the variance in relative abundance because the three groups of falcons were qualitatively different from one another in Salmonella load. There were 16 samples in which we detected no Salmonella reads and those were categorized as Salmonella negative. The relative abundances of Salmonella in the high group (7.4e-4, 7.9e-4, 1.8e-3, 2.8e-2, 2.2e-1) were generally an order of magnitude higher than those in the low groups (6.1e-5, 8.6e-5, 1.0e-4, 1.1e-4, 1.1e-4, 2.3e-4, 2.9e-4, 3.4e-4). Since there was a clear distinction in Salmonella load between these three groups, the variance was not considered. We have added a figure panel to demonstrate the relative abundances of Salmonella across these three groups in Figure 4C.

Line 243: first time it was mentioned that birds were in medical clinics.

Author’s response: All falcons in this study were healthy. In the UAE, certain veterinarian clinics provide boarding services for falcons. Owners house their falcons in these boarding facilities. We have added a statement in lines 258-261 of the revised manuscript to clarify this point.

Line413-415: Correlation of Beta with Alpha-diversity data is uncommon. Isn’t it a correlation of the same data set. Also Beta diversity measures consider species abundance. I would remove that analysis.

Author’s response: We differ with the Reviewer on this point. While both alpha and beta diversity are calculated using the same data (read counts), they measure two different aspects of the gut microbiota. Alpha diversity measures how diverse the gut microbiota is and beta diversity measures the distance between individuals based on their gut microbial species. If the species are common across individuals, we can expect diversity and composition to be correlated but if individuals have few species that are common, then the two metrics would not be correlated. In our falcons, we find a strong correlation between alpha and beta diversity measures, indicating that individual variance in their gut microbiome is low. We observe this in Figure 3. Furthermore, previous studies have performed such analyses. For example, Manor et al., (2020) and show human gut microbiome composition varies along the diversity gradient.

Line 430: it would be good to give the relative abundances here for high and low load that the reader gets directly an impression on the key abundance of Salmonella

It would be interning which genera were the other most abundant once, no discussion is given on them.

Author’s response: We have added a figure panel to demonstrate the relative abundances of Salmonella across these three groups in Figure 4C. We have also described the most abundant genera in the Discussion section in lines 554-559 of the revised manuscript.

Line 442: no -> not detected

Author’s response: Changed in line 472 of the revised manuscript.

Lines 444-446: How well is this prediction (% similarity values available?)

Author’s response: The predictions were made using a computational package called PICRUST2, which is specifically designed to predict the gene contents of environmental samples using 16S rRNA ASVs by aligning them to a comprehensive reference of bacterial and archaeal genomes. The functional pathways predicted by PICRUSt2 is comparable to pathways identified using shotgun MGS and therefore, it has been widely implemented in 16S rRNA studies to gain functional insights. Unfortunately, % similarity values are not available for the predictions from PICRUST2.

Lines 442-448: I am generally careful for those predictions. How precise are those correlations.

Have the authors also checked for the abundance of other pathogens?

Author’s response: We agree with the Reviewer that these are computational predictions. Recognizing this, we explicitly specify these are “predicted pathways” and the differences in these pathways may represent “potentially” altered the gut microbial functions. We are grateful to the Reviewer for raising this point. We have also added a statement describing this in the Discission section in lines 602-607 of the revised manuscript.

I would prefer a list of ASVs assigned to specific pathogens. Was really nothing detected?

It is clear that those data are just hinds, but never the less, interesting to give, 

Author’s response: We describe the most abundant genera in the falcons and ASVs associated with Salmonella in S7 Table. We also discuss the limitations of 16S rRNA data in detecting pathogenic bacteria in our dataset in the Discussion Section in lines 560-563 of the revised manuscript.

Line 477: “Stark” is that really English or German?- better “strong”?

Author’s response: We have changed “stark” to “strong” in line 508 of the revised manuscript.

Lines 536-540: however a list of taxa with potential pathogens can be given and discussed

Author’s response: We have included these ASVs in S7 Table.

I would consider more details in the discussion, especially to the limited quality of information obtained by the amplicon approach:

Author’s response: We discuss the limitations of 16S rRNA data in detecting pathogenic bacteria in our dataset in the Discussion Section first part is answered in comment about lines 444-446 (previous comment). These predictions are not coming from correlations in our paper but from literature search for the predicted pathways. We searched for other pathogenic genera that are not known commensal members of gut microbiota. We did not find any other potential pathogens at appreciable levels. This is clarified now in lines 560-561 and 596-598 of the revised manuscript.

- When the Salmonella load is discussed I would check a consistent colonization. Each falcon was analyzed just once. Must not be done in this paper, but should be discussed.

Author’s response: We thank the Reviewer for this insightful point. We have added a statement discussing this point in the Discussion section of the revised manuscript in lines 598-599.

- It was not discussed if the food/food source may be the reason for a Salmonella contamination. For readers as me which are not Falcon experts it would be interesting to get more information of the nutrition and food sources in the country of analysis. I would consider a clear link to food source. It the food quality controlled? Can that be the primary source for Salmonella infections…. All questions the paper opens to me. To get respective information in the paper will enhance the quality.

Author’s response: We agree with the Reviewer that determining the source of Salmonella in falcons would be interesting. Since Salmonella is a foodborne pathogen, infected meat is the most probable source. The birds could have acquired it from contaminated food in the housing facilities but they could also have been obtained from the wild when the falcons are taken for hunting excursions. Future strain-level metagenomic sequencing may enable distinction between wild versus commercial Salmonella strains. We have included this point in lines 560-569 of the revised manuscript.

- Amplicon data are all just relative abundance data, I would recommend to perform a quantitative detection of the Salmonella load (infA qPCR works for example well). If DNA samples are still in a freezer that can be done quickly and can be added to the paper. Otherwise again it should be at least discussed as important control in subsequent studies.

Author’s response: This is a valid suggestion from the reviewer. However, obtaining the qPCR reagents and protocol optimization can take months, which will substantially delay the publication of this manuscript. Although we are unable to incorporate this into the current manuscript, we fully intend to integrate it in our future studies. We have included this point in the discussion section of the revised manuscript in lines 598-602.

- Cultivation of respective Salmonella should be recommended. This again would enhance the knowledge in the growth state of Salmonella, alive, active -> culturable, dead or VBNC? Again should be at least discussed. I think there are respective studies available which worked on that.

Author’s response: A valid point by the Reviewer again. We have discussed this in lines 567-569 and 605-607. 

Reviewer #2: The manuscript by Ahmad and colleagues focuses on falcon microbiota. The aim of the manuscript is very interesting and fills the gap in knowledge about wild birds’ microbiota. However, the manuscript is confusing, it supposed to be focusing on falcons’ microbiota but spend a lot of time on reanalyzing somebody else data. The authors need to decide what they want to accomplish: provide comprehensive analysis of falcon microbiota or perform meta-analysis of birds’ microbiota. I think the former would be more valuable to scientific community.

Author’s response: We thank the reviewer for careful analysis of our manuscript. However, we differ from the Reviewer’s point of view. The overall goal of our research was to perform a comparative analysis of the falcon gut microbiota in the context of avian evolution. The specific hypothesis we were testing was that diet rather than the phylogenetic relationship has a larger effect on the falcon gut microbiome. By analyzing the previously published dataset from Song et al., which consists of gut microbiota from diverse species, we were able to detect a small but detectable effect of phylogeny on bird gut microbiota as shown in Fig 1. Next, we used this same dataset to show that diet has a much larger contribution in the gut microbiota of birds, which is shown in Fig 2. Both of these are novel findings from a pre-existing dataset. Finally, we integrated our falcon data with the Song et al. dataset to assess the effect of diet versus phylogeny and found that diet trumps phylogeny in falcons, which is shown in Fig 3. These are novel findings from our research. Importantly, without careful and detailed analysis of Song et al. dataset, we would not be able to determine the effect of diet on the falcon gut microbiota. If we were to focus only on our falcon data without integrating the Song et al. dataset, it would result in a descriptive study which would be less valuable to the scientific community. Recognizing the Reviewer’s point, we have moved parts of this section to figure legends, which has shortened this section and reduced redundancy.

Additional comments:

1. Microbiome refers to microbiota, their function, metabolome, etc. The paper only focusses on composition and predicted function; therefore, microbiota should be used instead microbiome.

Author’s response: We have changed “microbiome” to “microbiota” throughout the revised manuscript.

2. Line 76-81: this should be included in Result section when the metanalysis is described or in discussion section.

Author’s response: As per Reviewer’s request, we have moved parts of this section to the Results (lines 437-440) and Discussion (lines 607-608) sections of the revised manuscript. Integrating this comment has improved the flow of the manuscript and we thank the Reviewer for this suggestion.

3. Line 117, 260-280 ratio close to 1 indicates poor DNA quality. Have you performed any other quality check?

Author’s response: We thank the Reviewer for raising this point. We are confident that the DNA was of high quality for several reasons. First, we reported the range of 260/280 ratios from our samples. There was only 1 sample with the 260-280 ratio of 1.04, all others were near 1.8, indicating high quality DNA. Second, we were able to amplify DNA from all of the samples and the quality of the amplified product was measured using several techniques described in lines 132-138 of the revised manuscript. Third, all of the samples produced high quality libraries and yielded high sequencing depth. Finally, none of these samples were clear outliers in any of the subsequent analyses, indicating that variations in DNA quality did not play a role in this study. Thus, we have removed the DNA concentrations values in the revised manuscript.

4. Line 119-130: have the library been indexed? What was the sequencing length?

Author’s response: We thank the Reviewer for these points. The libraries were indexed and multiplexed sequencing was performed to generate reads of 250 bp length. We have included this description in lines 138-141 of the revised manuscript.

5. Line 248: new 16S analysis workflow was created, why? Why not follow already established workflow for 16S analysis? What is the advantage of the new workflow?

Author’s response: We optimized an existing 16S rRNA analysis workflow presented by Callahan et al. (2016), which is widely used in microbiome studies. We improved this workflow in multiple ways: (1) we optimized the parameters to analyze the data used in this study; (2) added additional analyses modules including the random forest classifier and PICRUST2 analysis; and (3) included additional R packages for data visualization. We have made the sequencing data and associated metadata along with the entire workflow publicly available as a supplementary files (S1 and S2 files). We hope these will be useful resource to future avian microbiome studies. We have amended the sentence in lines 266-268 in the revised manuscript to clarify the Reviewer’s point.

---

## [Decision Letter · Decision Letter 1]

23 Oct 2023

Falcon gut microbiota is shaped by diet and enriched in Salmonella

PONE-D-23-18286R1

Dear Dr. Jha,

We’re pleased to inform you that your manuscript has been judged scientifically suitable for publication and will be formally accepted for publication once it meets all outstanding technical requirements.

Kind regards,

Christopher Adenyo, Ph.D.

Academic Editor

PLOS ONE

Additional Editor Comments (optional):

Reviewers' comments:

Reviewer's Responses to Questions

**Comments to the Author**

1. If the authors have adequately addressed your comments raised in a previous round of review and you feel that this manuscript is now acceptable for publication, you may indicate that here to bypass the “Comments to the Author” section, enter your conflict of interest statement in the “Confidential to Editor” section, and submit your "Accept" recommendation.

Reviewer #2: All comments have been addressed

2. Is the manuscript technically sound, and do the data support the conclusions?

Reviewer #2: Yes

3. Has the statistical analysis been performed appropriately and rigorously? 

Reviewer #2: Yes

4. Have the authors made all data underlying the findings in their manuscript fully available?

Reviewer #2: Yes

5. Is the manuscript presented in an intelligible fashion and written in standard English?

Reviewer #2: Yes

6. Review Comments to the Author

Reviewer #2: The authors' responses are acceptable. The manuscript has been improved.

7. PLOS authors have the option to publish the peer review history of their article (what does this mean?). If published, this will include your full peer review and any attached files.

Reviewer #2: No

---

## [Editor Report · Acceptance letter]

3 Nov 2023

PONE-D-23-18286R1 

Falcon gut microbiota is shaped by diet and enriched in *Salmonella*

Dear Dr. Jha:

I'm pleased to inform you that your manuscript has been deemed suitable for publication in PLOS ONE. Congratulations! Your manuscript is now with our production department. 

Kind regards, 

on behalf of

Dr. Christopher Adenyo 

Academic Editor

PLOS ONE